# Instruments used to assess gender-affirming healthcare access: A scoping review

**Seán Kearns** [1,2]*, **Philip Hardie**[3‡], **Donal O'Shea**[1,2‡], **Karl Neff**[1,2]

**1** School of Medicine, University College Dublin, Dublin, Ireland, **2** St Columcille's Hospital, Dublin, Ireland, **3** Nursing Programme, Hibernia College, Dublin, Ireland

☯ These authors contributed equally to this work.
‡ PH and DO also contributed equally to this work.
* Sean.kearns1@ucdconnect.ie

## Abstract

### Purpose

The overall aim of this scoping review was to identify, explore and map the existing literature pertaining to healthcare access for transgender and non-binary individuals.

### Design

The scoping review followed Arksey and O'Malley's methodological framework, and the reporting adhered to the guidelines provided by the PRISMA Extension for Scoping Reviews.

### Methods

To gather relevant articles, a comprehensive search strategy was employed across four electronic databases, with the assistance of a university librarian. In addition, manual and internet searches were conducted for grey literature. From the initial search, a pool of 2,452 potentially relevant articles was retrieved, which was supplemented by an additional 23 articles from the supplemental search. After an independent review by two researchers, 93 articles were assessed, resulting in the inclusion of 41 articles in the review.

### Results

The literature highlights the identification of barriers and enablers, spanning across 32 individual data sets that affect healthcare accessibility for transgender and non-binary individuals. Leveque's five dimensions of healthcare access, namely approachability, acceptability, availability and accommodation, affordability, and appropriateness, were utilized to categorise these 42 factors. Some of the key themes that emerged in these dimensions include challenges in accessing information about services, concerns about acceptance from family and peers, past experiences of discrimination in healthcare settings, considerations related to cost and insurance, and the difficulty in finding appropriately trained competent providers.

**Data Availability Statement:** All relevant data are within the manuscript and its Supporting Information files.

**Funding:** The author(s) received no specific funding for this work.

**Competing interests:** The authors have declared that no competing interests exist.

## Conclusions

The review focused on the most commonly researched aspects of healthcare access and identified gaps in research and opportunities for future studies. The findings provide recommendations for policy and practice, which could guide the development of interventions aimed at addressing the barriers faced by transgender individuals seeking gender-affirming care.

## 1. Introduction

Transgender and non-binary individuals face significant challenges in accessing gender-affirming healthcare. Studies have consistently shown that this population experiences higher rates of discrimination and stigma in healthcare settings, which can result in decreased access to care and adverse health outcomes [1–3].

Furthermore, many healthcare providers lack knowledge of and training in transgender healthcare, which can result in inadequate or inappropriate care [4]. This is particularly concerning for transgender and non-binary youth, who may be at particular risk for adverse outcomes such as depression, anxiety, and suicide [5].

Addressing these barriers to healthcare access is critical in promoting the health and wellbeing of transgender and non-binary individuals. The care provided should be tailored to each individual's specific needs, and will often include medical interventions such as hormone blockers, cross-sex hormones, and surgical procedures [6].

Healthcare navigation can be a significant challenge for transgender individuals and this is evident in media representation and through research. Therefore, this study aims to explore access to gender-related healthcare, particularly how healthcare is assessed in this population, by using Levesque's healthcare access theory [7]. The choice of Levesque's healthcare htheory is justified in this study because it takes into account the complex and multifaceted nature of healthcare access, encompassing various factors at both individual and system levels. It recognises that healthcare access is not only influenced by the availability and affordability of healthcare services but also by individual characteristics, such as socioeconomic status, cultural beliefs, and health literacy. Furthermore, it acknowledges that healthcare access is not a static phenomenon and may change over time due to various factors such as policy changes, social norms, and healthcare delivery models.

This theory has been used in previous studies examining healthcare access for marginalized populations, including transgender and non-binary individuals [8, 9], and provides a useful framework for identifying and addressing barriers to healthcare access in this population. By applying this theory to a scoping review of healthcare access instruments for transgender individuals, the study could identify gaps in existing measures and inform the development of more comprehensive and inclusive instruments.

A scoping review is the most appropriate type of review in this case because it allows for a comprehensive exploration of the existing literature, which is essential in identifying the range of healthcare access instruments used and the influencing factors on access in the transgender population. Ultimately, this scoping review can contribute to the development of more effective strategies to promote equitable access to gender-affirming healthcare for transgender individuals.

The overall aim of this scoping review is to identify, explore and map the existing literature pertaining to healthcare access for transgender and non-binary individuals.

Specifically, the objectives of this scoping review are to:

- Determine the types of instruments or tools used to evaluate healthcare access among transgender and non-binary individuals seeking gender-affirming care.

- Compare the geographical distribution of these assessment tools.

- Identify the theories that underpin the research in this field.

- Determine the most commonly reported challenges faced by transgender and non-binary individuals when navigating healthcare systems.

- Describe the standard methodological approaches utilized in the research, including study design and outcomes.

- Identify any gaps or areas where further research is needed.

This scoping review will systematically compare quantitative instruments employed in the assessment of healthcare access among transgender and non-binary individuals, establishing itself as the first of its kind. This distinctive approach holds significance for several reasons: firstly, it offers a comprehensive overview of key factors influencing healthcare access; secondly, it applies an established healthcare access theory to this specific population; and thirdly, it compares the factors influencing healthcare access in adult and youth cohorts, providing valuable insights into diverse perspectives and experiences.

## 2. Methods

### 2.1 Review design

This scoping review was conducted using the methodology described by Arksey and O'Malley (2005) [10], which was further improved by Levac et al. (2010) [11] and the Joanna Briggs Institute (2015) [12]. The study followed the Preferred Reporting Items for Systematic Reviews and Meta-Analysis extension for Scoping Reviews (PRISMA-ScR) recommended by Tricco et al. (2018) [13]. The scoping review protocol was developed and reviewed by healthcare access experts based in Ireland, as described in Kearns et al. (2023) [14].

### 2.2 Identification of research question

The first stage of this research involved identifying the research question through an initial literature review and consultation with clinicians and an expert panel of transgender and non-binary youth. The resulting research question was:

*What factors help and hinder access to gender-related healthcare and how are these factors identified by quantitative instruments?*

As researchers with backgrounds in providing transgender healthcare, we acknowledge that our experiences in clinical settings and academic environments have shaped our understanding of the challenges faced by transgender and non-binary individuals in accessing healthcare. Three of the authors actively work in providing transgender healthcare services, bringing a practical and applied perspective to the study. The final author, a nursing lecturer and module developer, contributes an academic viewpoint to the research.

Furthermore, we recognize the importance of our own identities, including some of us identifying as members of the LGBT community, in influencing our approach to this study.

These experiences contribute to our commitment to inclusivity and understanding the unique healthcare needs of transgender and non-binary individuals.

Regarding the expert panel consisting of youths, the participants were young adults aged between 18 and 30 years old. They were intentionally recruited through maximum variation selection to ensure diverse experiences in accessing services as both youth and adults. This decision was made to enrich the study with a range of perspectives and insights into the evolving challenges faced by transgender and non-binary individuals throughout different stages of life

## 2.3 Identification of relevant studies

The study employed a systematic search strategy that involved four electronic databases; PsychINFO, CINAHL, Medline, and Embase. The databases were selected based on their relevance to the topic under investigation. The search strategy was formed with the assistance of a University librarian, in accordance with the recommendation of McGowan et al. (2020) [15]. The search was completed for all available literature up until December 2022.

The study protocol [14] contains a sample search strategy and inclusion-exclusion criteria. The search terms were guided by the PCC (Population, Concept, Context) mnemonic, as outlined by Peters et al. (2020) [16], and synonyms relating to the PCC were identified and utilized. The identified population in this study were transgender and non-binary individuals seeking gender-affirming care and included search terms such as 'transgender' OR 'non-binary' OR 'gender dysphoria', the concept is a healthcare accessibility instrument and included search terms such as 'quantitative' OR 'survey' or 'questionnaire', and the context is gender-related healthcare access and included search terms such as ' healthcare' OR 'health' OR 'gender care' OR 'hormones'. Each strategy was modified to meet the database-specified requirements for MeSH headings, Boolean operators, and truncation markers.

In addition to the electronic database search, the study team conducted a hand-search of the reference lists of included studies. A grey literature search was also conducted to identify potentially overlooked studies that meet the inclusion criteria and because it is a requirement for rigour in scoping reviews. This search was completed using "Google Scholar" and web searches, and the first one hundred results from identified keywords were assessed for eligibility. The PRISMA Flowchart (Fig 1) details the results obtained from the electronic database and grey literature searches.

## 2.4 Study selection

Studies were screened using the Rayaan screening software, which facilitated the management of data by enabling the upload of studies. This software was particularly helpful in eliminating duplicates (n = 1338). Subsequently, title and abstract screening was carried out by two authors, SK and PH, with a third author, DOS, available to address any conflicts that arose. Initially, the two main reviewers conducted an independent screening of 50 articles against the study's inclusion and exclusion criteria, and inter-rater reliability was assessed.

The pilot of 50 studies demonstrated high inter-rater reliability, and consequently, 2,452 titles and abstracts were screened, resulting in the identification of 74 studies for full-text screening. No adaptations were required following the pilot. The full-text screening was conducted by SK, PH and KN, and 36 studies were included. The grey literature was evaluated by SK and KN through website searches, organizational reports, and reference lists of included studies, leading to the inclusion of 5 additional studies.

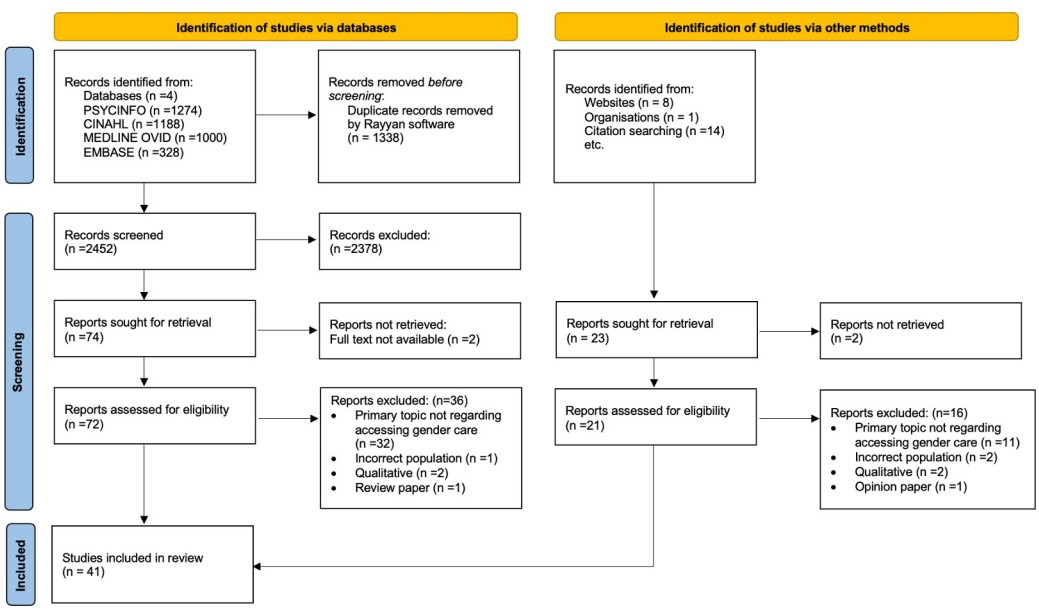

**Fig 1. PRISMA flow chart.**

### 2.5 Charting the data

An Excel sheet was utilized to extract data for the study, with agreed-upon headings that matched the research question and objectives (refer to Table 1). The data was not evaluated for quality, which is typical for scoping review methodology. Nevertheless, the data was collated, and the overall findings were narratively described in the results section. These results were shared through a in-person group discussion with a pre-existing expert panel consisting of transgender and non-binary youths who confirmed that the results were consistent with their personal experiences.

## 3. Results: Collating, summarizing and reporting

Results from the database and grey literature search are show in the PRISMA diagram below (Fig 1). A final list of 41 records met all the eligibility criteria.

### 3.1 Study characteristics

The review includes a total of 41 records that represent research from 18 countries across 5 continents, namely Asia, Australia, Europe, North America, and South America. The United States has the highest number of publications in this field (n = 22), followed by Canada (n = 6), Europe (n = 5), Asia (n = 4), Australia and New Zealand (n = 2), and South America (n = 2). The European studies represent nine countries, while Asia and South American studies represent three and two countries, respectively. The studies were conducted in twelve high-income countries, five upper-middle-income countries, and one lower-middle-income country.

These 41 records represent 32 individual data sets, with five studies sharing a data set from the United States [17–21], two studies sharing a data set from Germany [22, 23], and two studies sharing a data set from Korea [24, 25]. Half of the studies were published in the last four years (n = 21), while the earliest study was conducted in 2008 (n = 1). Fig 2 shows the number of included studies per year. All of the 41 studies utilized cross-sectional study designs. A majority of these studies were conducted online (n = 27/41), while six studies recruited participants from clinics, three studies recruited in-person, and four employed a combination of

**Table 1. Included studies characteristics.**

| Author and year, Study title, Study aim | Country and/or region | Population, Sample Size, Sample age | Methods | Access factors assessed | Overall findings | Instrument(s) used | Instrument(s) development | Inclusion of PPI |
|---|---|---|---|---|---|---|---|---|
| (Bakko et al., 2020) Study title: Transgender-Related Insurance Denials as Barriers to Transgender Healthcare: Differences in Experience by Insurance Type Study aim: This study investigates the association between transgender and non-binary individuals' experiences of different forms of transgender-related insurance denials and insurance type. | USA | Trans/NB (n = 27,715) Age: 18+ | • Cross-sectional survey • Online | • Gender identity considerations • Health insurance coverage/denial | • Models revealed significant relationships between transgender-related insurance denials and insurance type for 11,320 transgender and non-binary adults. • Compared with those with private insurance, Medicaid coverage was associated with an increased likelihood of experiencing denials for hormone therapy; having no in-network surgery providers was associated with Medicare or Medic- aid; and military-based insurance was associated with transition-related surgery denials. | 2015 United States Transgender Survey (a sub section of overall findings) | The USTS survey instrument was developed over the course of a year by a core team of researchers and advocates in collaboration with dozens of individuals with lived experience, advocacy and research experience, and subject-matter expertise. Exact details on manner of PPI not specified but mentioned. A pilot study of 100 individuals influenced the survey too. | Yes |
| (Bradford et al., 2013) Study title: Experiences of Transgender-Related Discrimination and Implications for Health: Results From the Virginia Transgender Health Initiative Study Study aim: To examine relationships between social determinants of health and experiences of transgender-related discrimination reported by transgender people in Virginia. | USA, Virginia | Trans/NB (n = 350) Age:18+ | • Cross-sectional survey • Online | • Experienced discrimination in healthcare • Gender identity considerations • Family/peers not supportive • Finding a provider/service • Health insurance coverage/denial | • Of participants, 41% (n = 143) reported experiences of transgender related discrimination. • Factors associated with transgender-related discrimination were geographic context, gender (female-to male spectrum vs male-to-female spectrum), low socioeconomic status, being a racial/ethnic minority, not having health insurance, gender transition indicators (younger age at first transgender awareness), health care needed but unable to be obtained (hormone therapy and mental health services), history of violence (sexual and physical), substance use health behaviours (tobacco and alcohol), and interpersonal factors (family support and community connectedness). | The Virginia Transgender Health Initiative Study | A Transgender Taskforce was formed to involve transgender individuals at all levels of study design and implementation. More exact details on inclusion were not provided in the paper. | Yes |

*(Continued)*

Table 1. (Continued)

| Author and year, Study title, Study aim | Country and/or region | Population, Sample Size, Sample age | Methods | Access factors assessed | Overall findings | Instrument(s) used | Instrument(s) development | Inclusion of PPI |
|---|---|---|---|---|---|---|---|---|
| (Bretherton et al., 2021) Study title: The Health and Well-Being of Transgender Australians: A National Community Survey Study aim: To better understand the health status and needs of Australian trans people to guide resources and health and well-being programs. | Australia | Trans/NB (n = 928) Age:18+ | • Cross-sectional survey • Online | • Lack of service information • Pathways difficult to navigate • Experienced discrimination in healthcare • Finding a provider/ service • Financial cost • Mental health assessment requirement | • Discrimination in accessing health care was reported by 26% and verbal abuse and physical assault were reported by 63% and 22%, respectively. • 59% of participants faced challenges in accessing hormonal treatment. Factors impacting this included: difficulty finding a doctor, financial factors and difficulty navigating pathways. • Views were elicited on the if mental assessments should be a requirement, 74% felt they should in most cases. | Unnamed survey. Designed by research team | No specifics in paper on PPI contribution to project. | No |
| (Burgwal & Motmans 2020) Study title: Trans and gender diverse people's experiences and evaluations with general and trans-specific healthcare services: a cross-sectional survey Study aim: This research aims to analyse access to, and experiences with, trans-specific healthcare services, as well as the evaluation of trans-specific healthcare by trans respondents, taking different socio-demographic variables and gender identities into account. | Georgia, Poland, Serbia, Spain, and Sweden | Trans/ gender-diverse (n = 742) Age: 16+ | • Cross-sectional survey • Online | • Lack of service information • Pathways difficult to navigate • Experienced discrimination in healthcare • Gender identity considerations • Fear of seeking care • Unsure if want/need medical intervention • Fertility/child related considerations • Fear of discrimination in healthcare • Finding a provider/ service • Waiting times • Financial cost • Health insurance coverage/denial • Expertise and skill of provider (Need to teach) • Confidence in service available | • Trans and gender diverse individuals were presented as two cohorts in this study. • 73.6% of trans participants had sought medical/ psychological support for being trans compared to 40% of gender diverse participants. • Trans participants reported motivations for not accessing trans-specific care due to being afraid, lack of confidence in services and fear of prejudice from providers. 31% reported that they did not want/need help. • Gender diverse participants reported motivations for not accessing trans-specific care due to not having confidence in services, not knowing where to go and fear of prejudice from providers. 51% reported that they did not want/need help. | Trans Health Study | The questionnaire development and data collection was coordinated by Transgender Europe (TGEU) using the framework of their project "Trans Health Study", in close collaboration with its partner organizations "Women's Initiative Supportive Group" (WISG) (Georgia), "Trans-Fuzja" (Poland), "Daniela Fundación" (Spain), "Geten" (Serbia), and the "Riksförbundet för homosexuellas, bisexuellas, transpersoners, queeras och intersexpersoners rättigheter" (RFSL) (Sweden). No specifics in paper on PPI contribution to project. | No |

(Continued)

Table 1. (Continued)

| Author and year, Study title, Study aim | Country and/or region | Population, Sample Size, Sample age | Methods | Access factors assessed | Overall findings | Instrument(s) used | Instrument(s) development | Inclusion of PPI |
|---|---|---|---|---|---|---|---|---|
| (Clark et al., 2018) Study title: Non-binary youth: Access to gender-affirming primary health care Study aim: To document differences in access to gender-affirming health care between binary and non-binary identified trans youth and explore ways of meeting the health needs of non-binary youth within primary care settings. | Canada | Trans/ genderqueer (n = 839) Age:14–25 | • Cross-sectional survey • Online | • Lack of service information • Gender identity considerations • Family/peers not supportive • Unsure if want/need medical intervention • Hoped feeling would go away • Finding a provider/ service • Financial cost | • The statistical model used predicted that non-binary youth were twice as likely to experience barriers to accessing hormones than binary youth. • The barrier that most affected both groups was finding a doctor willing to prescribe hormones. • Lack of support from family was another major barrier. | Canadian Trans Health Survey | Developed drawing on questions from existing population health surveys suit- able for adolescent (e.g., British Columbia Adolescent Health Survey) and young adult (e.g., Canadian Community Health Survey) populations. No specifics in paper on PPI contribution to project. | No |
| (Costa et al., 2018) Study title: Healthcare Needs of and Access Barriers for Brazilian Transgender and Gender Diverse People Study aim: To explore transgender and gender diverse peoples (TGD) specific healthcare needs and struggles with access barriers in the Brazilian context. | Brazil | Trans/NB (n = 626) Age:18+ | • Cross-sectional survey • Online | • Lack of service information • Pathways difficult to navigate • Experienced discrimination in healthcare • Gender identity considerations • Fear of seeking care • Unsure if want/need medical intervention • Finding a provider/ service • Denied care • Financial cost • Expertise and skill of provider (Need to teach) • Comfort with provider | • A history of discrimination was associated with a 6.72 increase in healthcare avoidance. • Regarding accessing hormones, transwomen mainly sourced from internet pharmacies (39.2%), while transmen and gender diverse participants mainly sourced from medical specialists (53.2% and 33.3%). • For those who have yet to source hormones, transwomen and gender diverse participants noted it was because they were still deciding if hormones were right for them (41.1% and 60.0% respectively). • Transmen found it difficult it difficult to find a doctor who would prescribe hormones (55.4%). | Adapted from Trans PULSE study. | The original project created a Community Engagement Team (CET) of trans community members, selected through a province-wide application process. The 16 members represented, as much as conceivable, Ontario's diversity in terms of geography, newcomer status, age, ethno-racial groups, and trans identities. The CET provided valuable input, knowledge, and advice for numerous aspects of this project including survey and interview guide development; outreach and promotion of Trans PULSE; a 'knowledge to action' strategy for social change; as well as providing access to trans networks in diverse geographic areas. The Brazilian paper does not mention their own inclusion of PPI in their roll out. | Yes |

*(Continued)*

Table 1. (Continued)

| Author and year, Study title, Study aim | Country and/or region | Population, Sample Size, Sample age | Methods | Access factors assessed | Overall findings | Instrument(s) used | Instrument(s) development | Inclusion of PPI |
|---|---|---|---|---|---|---|---|---|
| (El-Hadi et al., 2018) Study title: Gender-Affirming Surgery for Transgender Individuals: Perceived Satisfaction and Barriers to Care Study aim: The purpose of this study was to examine the perceived satisfaction and barriers to care for transgender patients after they decide to undergo gender-affirming surgery (GAS). | Canada and USA | Trans/NB (n = 32) Age:18+ | • Cross-sectional survey • Online | • Lack of service information • Pathways difficult to navigate • Experienced discrimination in healthcare • Family/peers not supportive • Unsure if want/need medical intervention • Fertility/child related considerations • Finding a provider/service • Denied care • Financial cost • Health insurance coverage/denial • Expertise and skill of provider (Need to teach) | • The mean age of their first GAS was 33 years, and the range of wait time was 6 months to 7 years. • Most of the participants received information about GAS from transgender websites and transgender surgery clinics (91% and 50%, respectively). • Most participants (74%) felt like they had access to appropriate care and 89% felt like their surgeons provided enough information about GAS. • Participants stated several barriers toward receiving GAS: financial (73%), finding a physician (65%), and access to information (63%). • Surgical transition was important to the quality of life for 91% of participants and 100% were happy with their decision to undergo GAS. | Unnamed survey. Designed by research team | The survey was designed by the research team following a literature review. No specifics in paper on PPI contribution to project. | No |
| (Esseyl et al, 2017) Study title: Needs and concerns of transgender individuals regarding interdisciplinary transgender healthcare: A non-clinical online survey Study aim: To investigate the needs and concerns transgender (short: trans) individuals have concerning trans healthcare (THC) in interdisciplinary THC centres. | Germany, Hamburg | Trans/NB (n = 425) Age: 16+ | • Cross-sectional survey • Online | • Surgical follow up • Mental health assessment requirement • Model of care (centralised MDT vs decentralised) • Pre intervention counselling • Communication and contact with service • Regular contact with a set person • Integration of peer support programmes • Commitment to research • Involvement in decision making | • 96.5% of participants would like high decision making power connected with treatment associated decisions. • 94.7% of participants were in favour of THC centres establishing a peer-peer support programme with former patients supporting current patients. • Most participants were interested in trans specific healthcare to be provided at a THC but acknowledged fears associated with this model too. Especially regarding expectation from healthcare providers. | Unnamed survey. Designed by research team | The survey was designed by the research team in consultation with a working group consisting of trans representatives and healthcare providers. | Yes |

(Continued)

Table 1. (Continued)

| Author and year, Study title, Study aim | Country and/or region | Population, Sample Size, Sample age | Methods | Access factors assessed | Overall findings | Instrument(s) used | Instrument(s) development | Inclusion of PPI |
|---|---|---|---|---|---|---|---|---|
| (Ettner et al., 2016) Study title: Choosing a Surgeon: An Exploratory Study of Factors Influencing Selection of a Gender Affirmation Surgeon Study aim: To assess the importance attached to various factors involved in selecting a surgeon to perform gender affirmation surgery (GAS). | USA | Trans women (n = 54) Age:16+ | • Cross-sectional survey • In clinic | • Lack of service information • Pathways difficult to navigate • If provider is trans themselves • Expertise and skill of provider (Need to teach) • Comfort with provider | • Skill of provider most salient factor in decision-making. • Pragmatic concerns were next most important. • Personal characteristics were of less importance. • Transwomen were more likely to choose surgeon based on advice from trusted medical or mental health provider. | Unnamed survey. Designed by research team. | The questionnaire was an adaptation of the Lerman Perceived Involvement in Care Scale. No specifics in paper on PPI contribution to project. | No |
| (Feldman et al., 2021) Study title: Health and health care access in the US transgender population health (TransPop) survey Study aim: To describe and compare measures of health and health access among trans-gender, nonbinary, and cisgender participants | USA | Trans/NB (n = 274) Age:18+ | • Cross-sectional survey • Online | • Gender identity considerations • Fear of seeking care • Finding a provider/service • Financial cost • Health insurance coverage/denial • Expertise and skill of provider (Need to teach) • Model of care (centralised MDT vs decentralised) | • 90.5% of transgender participants had insurance but many participants still associated healthcare avoidance with cost. • Only 55.9% of participants had a transgender-related healthcare provider with non-binary individuals having significantly less. • 82.4% of overall participants would like access to a LGBT/transgender clinic or provider in the future. | The ""TransPop"" Survey | This questionnaire was a population based national questionnaire. No specifics in paper on PPI contribution to project. | No |
| (Heard et al.,2018) Study title: Gender dysphoria assessment and action for youth: Review of health care services and experiences of trans youth in Manitoba Study aim: To describe the paediatric transgender population accessing health care through the Manitoba Gender Dysphoria Assessment and Action for Youth (GDAAY) program, and report youth's experiences accessing health care in Manitoba. | Canada, Manitoba | Trans/NB (n = 25) Age:14–17 | • Cross-sectional survey • Online | • Experienced discrimination in healthcare • Fear of seeking care • Not out • Denied care • Waiting times • Expertise and skill of provider (Need to teach) • Comfort with provider | • Survey participants were aware of trans identity from average of 8.7 years but did not seek transition related health care until average of 13.3 years old. • Participants felt a strong onus of needing to educate their healthcare provider (70%). • Youth were worried about talking to healthcare providers as they were concerned as to the questions they would be asked, the lack of support they would get and felt services were not supportive of LGBT health. | Unnamed survey. Designed by research team | The questionnaire was developed from three previously validated surveys. No specifics in paper on PPI contribution to project. | No |

(Continued)

Table 1. (Continued)

| Author and year, Study title, Study aim | Country and/or region | Population, Sample Size, Sample age | Methods | Access factors assessed | Overall findings | Instrument(s) used | Instrument(s) development | Inclusion of PPI |
|---|---|---|---|---|---|---|---|---|
| (Hughto et al., 2017) Study title: Barriers to Gender Transition-Related Healthcare: Identifying Underserved Transgender Adults in Massachusetts Study aim: The present study sought to examine whether individual (e.g., age, gender), interpersonal (e.g., healthcare provider discrimination), and structural (e.g., lack of insurance coverage) factors are associated with access to transition-related care in a state-wide sample of transgender adults. | USA, Massachusetts | Trans/nb (n = 364) Age:18+ | • Cross-sectional survey • Online | • Experienced discrimination in healthcare • Age • Established social transition • Finding a provider/ service • Denied care • Financial cost • Health insurance coverage/denial • Expertise and skill of provider (Need to teach) • Mental health assessment requirement | • Younger age, being visually gender conforming, and having presented as transgender are seen to be protective factors in accessing transition related care. • Having a lower educational status and income, as well as restricted insurance is linked as barriers to care. • Previous negative experiences also impact access to care. | Unnamed survey. Designed by research team | Developed by the research team. Little information on how. No specifics in paper on PPI contribution to project. | No |
| (Johns et al., 2017) Study title: Socio-demographic factors associated with trans*female youth's access to health care in the San Francisco Bay Area Study aim: To test associations between socio-demographic variables and barriers to gender identity-based medical and mental health care. | USA, San Francisco | Trans women (n = 314) Age:16–24 | • Cross-sectional survey • Online | • Homelessness | • Having a history of unstable housing was associated with significantly higher odds of problems accessing both medical care (OR 2.16, 95% CI 1.12, 4.13) and mental health care due to gender identity (OR 2.65, 95% CI 1.08, 6.45). • Conversely, identifying as genderqueer/genderfluid, Latina, or living in dependent housing was associated with access to either medical or mental health care. | Unnamed survey. Designed by research team | Developed by the research team. Little information on how. No specifics in paper on PPI contribution to project. | No |

(Continued)

Table 1. (Continued)

| Author and year, Study title, Study aim | Country and/or region | Population, Sample Size, Sample age | Methods | Access factors assessed | Overall findings | Instrument(s) used | Instrument(s) development | Inclusion of PPI |
|---|---|---|---|---|---|---|---|---|
| (Kattari et al, 2020) Study title: Intersecting Experiences of Healthcare Denials Among Transgender and Nonbinary Patients Study aims: (1) Does the likelihood of being denied health care vary by gender within the TNB population, and if so, how? (2) How do socioeconomic and identity characteristics, including gender identity, race, income, disability status, age, and education level, affect the likelihood of TNB individuals being denied health care? | USA | Trans/NB (n = 27,715) Age: 18+ | • Cross-sectional survey • Online | • Gender identity considerations • Age • Denied care • Financial cost | • 8% of participants had been refused trans-related care. • 3% had been refused general health care. • Transgender (compared with nonbinary), older, biracial, or multiracial, and lower-income participants, as well as those with less than a high school diploma and those with disabilities, were significantly more likely to experience refusal of care in general or trans-specific healthcare settings. | 2015 United States Transgender Survey (a sub section of overall findings) | The USTS survey instrument was developed over the course of a year by a core team of researchers and advocates in collaboration with dozens of individuals with lived experience, advocacy and research experience, and subject-matter expertise. Exact details on manner of PPI not specified but mentioned. A pilot study of 100 individuals influenced the survey too. | Yes |
| (Kachen et al, 2020) Study title: Health Care Access and Utilization by Transgender Populations: A United States Transgender Survey Study Study aim: To elucidate health disparities regarding access to and utilization of health care and experiences with discrimination within the transgender community. | USA | Trans/NB (n = 27,715) Age: 18+ | • Cross-sectional survey • Online | • Experienced discrimination in healthcare • Gender identity considerations • Finding a provider/ service • Denied care • Financial cost • Health insurance coverage/denial • Expertise and skill of provider (Need to teach) • Comfort with provider | • 84.3% of participants had health insurance. • Transmen and transwomen felt their providers had very good knowledge about medical issues. • TM were 1.29 times more likely to have a transgender-specific care provider than TF. • NB participants were 89% less likely to have a transgender specific health provider. | 2015 United States Transgender Survey (a sub section of overall findings) | The USTS survey instrument was developed over the course of a year by a core team of researchers and advocates in collaboration with dozens of individuals with lived experience, advocacy and research experience, and subject-matter expertise. Exact details on manner of PPI not specified but mentioned. A pilot study of 100 individuals influenced the survey too. | Yes |

(Continued)

Table 1. (Continued)

| Author and year, Study title, Study aim | Country and/or region | Population, Sample Size, Sample age | Methods | Access factors assessed | Overall findings | Instrument(s) used | Instrument(s) development | Inclusion of PPI |
|---|---|---|---|---|---|---|---|---|
| (Koehler et al., 2018) Search title: Genders and Individual Treatment Progress in (Non-)Binary Trans Individuals Search aims: To gain insight into the individual health care experiences and needs of binary and NBGQ individuals to improve their health care outcomes and experience. | Germany, Hamburg | Trans/NB (n = 415) Age: 16+ | • Cross-sectional survey • Online | • Gender identity considerations • Unsure if want/need medical intervention • Finding a provider/service | • Of those who accessed transition related care, binary participants reported a significantly larger percentage than NBGQ. • Binary and NBGQ differ in sociodemographic and planned and received treatments. • NBGQ healthcare users require less treatments to completion than binary. | Unnamed survey. Designed by research team | The survey was designed by the research team in consultation with a working group consisting of trans representatives and healthcare providers. | Yes |
| (Lee et al., 2018) Study title: Experiences of and barriers to transition-related healthcare among Korean transgender adults: focus on gender identity disorder diagnosis, hormone therapy, and sex reassignment surgery Study aim: This study aimed to investigate the experiences of and barriers to transition-related healthcare, including GID diagnosis, hormone therapy, and sex reassignment surgery, for transgender adults in Korea. | Korea | Trans/NB (n = 278) Age: 19+ | • Cross-sectional survey • Online | • Lack of service information • Gender identity considerations • Family/peers not supportive • Fear of seeking care • Unsure if want/need medical intervention • Fertility/child related considerations • Finding a provider/service • Denied care • Financial cost • Mental health assessment requirement | • 48% of those who have not received a GID diagnosis was due to "having financial difficulties". • 54.9% of those who delayed hormone therapy was due to "having financial difficulties". • The main reason participants purchased non-prescribed hormones was due to "not having a diagnosis from a psychiatrist (55.7%). • A small number of participants were denied care (5.3%) • 47.1% of participants feared accessing healthcare would impact their economic activities and 15.7% of participants had friends and family discourage seeking care. | Rainbow Connection Project II– Korean Transgender Adults' Health Study | A nationwide cross-sectional survey of a non-probability sample of Korean transgender adults. No specifics in paper on PPI contribution to project. | No |

*(Continued)*

**Table 1.** (Continued)

| Author and year, Study title, Study aim | Country and/or region | Population, Sample Size, Sample age | Methods | Access factors assessed | Overall findings | Instrument(s) used | Instrument(s) development | Inclusion of PPI |
|---|---|---|---|---|---|---|---|---|
| **(Lee et al., 2022) Study title: Does Discrimination Affect Whether Transgender People Avoid or Delay Healthcare?: A Nationwide Cross-sectional Survey in South Korea Study aim: To examine the association between perceived discrimination and healthcare avoidance and delay (HAD) among transgender adults in South Korea (hereafter Korea).** | Korea | Trans/NB (n = 244) Age: 19+ | • Cross-sectional survey • Online | • Experienced discrimination in healthcare • Gender identity considerations | • 37.7% of the sample had avoided healthcare due to discrimination in last twelve months. • Transmen were most likely to avoid healthcare following discrimination than their counterparts. • Transgender participants who experienced discrimination "only due to their transgender identity" and due to "their transgender identity and other reasons" reported a 1.91 and 1.96 fold higher prevalence of healthcare avoidance than those who never experienced discrimination. | Rainbow Connection Project II–Korean Transgender Adults' Health Study | A nationwide cross-sectional survey of a non-probability sample of Korean transgender adults. No specifics in paper on PPI contribution to project. | No |
| **(Lett et al., 2022) Study title: Ethnoracial inequities in access to gender affirming mental health care and psychological distress among transgender adults Study aim: To evaluate the impact of systemic racism on access to gender-affirming mental health care (GAMHC) among transgender people of colour (TPOC).** | USA | Trans/NB (n = 20,967) Age: 18+ | • Cross-sectional survey • Online | • Gender identity considerations • Ethnoracial considerations | • This study found decreased access to GAMHC(gender affirming mental health care) across all TPOC (transgender persons of colour) groups. • Inequities in access to GAMHC were most severe among assigned male at birth respondents in the Black/African-American group (aOR 0.51, 95% CI 0.37–0.71), Latino/a/e/Hispanic group (aOR 0.52, 95% CI 0.42–0.65), and Native American group (aOR 0.59, 95% CI 0.38–0.94). • Among all respondents, severe psychological distress was highest among Native American respondents (47.4%), Latino/a/e/Hispanic (47.1%) respondents, and other/multiracial respondents (46.7%) and lowest among whites (39.9%). • Further, among all TPOC, access to GAMHC was associated with decreased odds of severe psychological distress (aOR 0.74, 95% CI 0.62–0.87). | 2015 United States Transgender Survey (a sub section of overall findings) | The USTS survey instrument was developed over the course of a year by a core team of researchers and advocates in collaboration with dozens of individuals with lived experience, advocacy and research experience, and subject-matter expertise. Exact details on manner of PPI not specified but mentioned. A pilot study of 100 individuals influenced the survey too. | Yes |

(Continued)

**Table 1.** (Continued)

| Author and year, Study title, Study aim | Country and/or region | Population, Sample Size, Sample age | Methods | Access factors assessed | Overall findings | Instrument(s) used | Instrument(s) development | Inclusion of PPI |
|---|---|---|---|---|---|---|---|---|
| (Lozano-Verduzco & Melendez 2021) **Study title: Transgender individuals in Mexico: exploring characteristics and experiences of discrimination and violence** **Study aim: To explore characteristics of violence and discrimination among mexican transgender population** | Mexico, Mexico City | Trans/NB (n = 148) Age: 14+ | • Cross-sectional survey • Online and in person | • Experienced discrimination in healthcare | • Over 10% of participants reported having used hormones. Of those participants, less than 40% are currently using hormones. • Very few used silicone or surgeries as part of their gender-affirmation process. • Only 5.4% of participants reported that they had access to hormones through a health professional and 2.1% reported that they had access to them from another transgender person. • A small number of participants (N = 3) reported having had surgeries. • The frequency of discrimination varies from 9.4% (discriminated by the military or denied a health service) to over 36.5% (referred to psychological treatment under the premise that this treatment will revert the persons trans identity). | Unnamed survey. Designed by research team | Part of a larger LGBT study. Questionnaire designed by the research team. No specifics in paper on PPI contribution to project. | No |
| (Manzoor et al., 2022) **Study title: Health Problems & Barriers to Healthcare Services for the Transgender Community in Lahore, Pakistan** **Study aim: To explore the major health problems and barriers in getting health care by transgender community in Lahore, Pakistan** | Pakistan | Trans (n = 214) Age: 18+ | • Cross-sectional survey • In person | • Experienced discrimination in healthcare • Gender identity considerations • Family/peers not supportive • Feeling of shame • Finding a provider/ service • Denied care • Financial cost • Expertise and skill of provider (Need to teach) | • 78.5% of sample were transfeminine and 21.5% transmasculine. • The main reasons for not accessing quality care were: non acceptance (20.7%), feeling ashamed (28.7%), non-availability of national identity card (44.5%) and cost (6.1%). • Significant association of transgender female with consultation with doctors (p = 0.013), seeking care at government hospitals (p = 0.038) poor experience at health care facility (0.050), neglect during medical treatment (p = 0.015) and feeling of discrimination during treatment (p = 0.042). | Unnamed survey. Designed by research team | The research questionnaire was developed by the research team in partnership with NGO's. The team contacted transgender "gurus" in their area to endeavour to reach marginalised communities and provided additional health assessments in return for participation. Informed consent forms were adapted to local language Urdu. | Yes |

*(Continued)*

**Table 1.** (Continued)

| Author and year, Study title, Study aim | Country and/or region | Population, Sample Size, Sample age | Methods | Access factors assessed | Overall findings | Instrument(s) used | Instrument(s) development | Inclusion of PPI |
|---|---|---|---|---|---|---|---|---|
| (McNeil et al., 2013) **Study title: Speaking from the Margins, Trans Mental Health and Wellbeing in Ireland** Study aim: To examine mental health and wellbeing of trans people in Ireland | Ireland | Trans/NB (n = 164) Age: 18+ | • Cross-sectional survey • Online | • Experienced discrimination in healthcare • Unsure if want/need medical intervention • Finding a provider/ service • Denied care • Waiting times • Expertise and skill of provider (Need to teach) • Comfort with provider • Mental health assessment requirement • Confidence in service available | • 46% of participants were seen by a GIC. • 55% waited less than a year, 21% between one and two years, 25% more than 2 years. • 55% of participants felt they experienced difficulty in accessing a treatment they felt they needed from a GIC. | Speaking from the Margins | The questionnaire was developed in line with the Scottish Transgender Alliance survey carried out in the UK, which was based on the TransPulse survey in Canada. PPI was utilised and community members were consulted in the area. | Yes |
| (Nemoto et al., 2004) **Study title: Health and Social Services for Male-to-Female Transgender Persons of Colour in San Francisco** Study aim: 1. What are the health and social service needs of MtF transgender persons of colour in San Francisco? 2. How do MtF transgender persons of colour perceive the health and social services available to them? | USA, San Francisco | Trans women (n = 332) Age: 18+ | • Cross-sectional survey • In person | • Lack of service information • Experienced discrimination in healthcare • Ethnoracial considerations • Finding a provider/ service • Waiting times • Financial cost • Expertise and skill of provider (Need to teach) | • 91% of sample had a history of hormone use while 75% were using at time of study. • 59% had plans for future gender-related procedures. • Waiting lists are the most prohibitive factor to accessing gender clinics (34%). • This is followed by doctors being insensitive to transgender issues (22%) and lack of knowledge (21%). | Unnamed survey. Designed by research team | Individual survey interviews using transgender specific instruments for trans women only. No specifics in paper on PPI contribution to project. | No |

*(Continued)*

Table 1. (Continued)

| Author and year, Study title, Study aim | Country and/or region | Population, Sample Size, Sample age | Methods | Access factors assessed | Overall findings | Instrument(s) used | Instrument(s) development | Inclusion of PPI |
|---|---|---|---|---|---|---|---|---|
| (Newhook et al., 2018) Study title: The TransKidsNL Study: Healthcare and Support Needs of Transgender Children, Youth, and Families on the Island of Newfoundland Study aim: 1. To describe the healthcare and support needs of transgender and gender-questioning children and youth in NL, as understood by youth and by parents. 2. To examine the main concerns and hopes of transgender and gender-questioning youth and by parents in NL. | Canada, Newfoundland and Labrador | Trans/NB (n = 24) Age: 12–17 | • Cross-sectional survey • In clinic | • Lack of service information • Family/peers not supportive • Fear of seeking care • Unsure if want/need medical intervention • Waiting times • Expertise and skill of provider (Need to teach) • Mental health risks | • Only 13% of participants described their family as fully supportive. • Young people identified the correct name and pronoun use as the most important support needed from their parents (41.2%) followed by being listened to and believed (29.4%). | The TransKidsNL Study | The questionnaire was developed by a Trans Health Research Group and included participation from a local peer run support group for parents and guardians of transgender, two-spirit and gender diverse children and youth. | Yes |
| (Pitts et al., 2009) Study title: Transgender People in Australia and New Zealand: Health, Well-being and Access to Health Services Study aim: To examine the health and well-being of transgender people in Australia and New Zealand including healthcare access | Australia and New Zealand | Trans (n = 253) Age: 18+ | • Cross-sectional survey • Online | • Experienced discrimination in healthcare • Finding a provider/ service • Waiting times • Financial cost • Health insurance coverage/denial • Expertise and skill of provider (Need to teach) • Comfort with provider | • 50.7% of Australian respondents had private health insurance compared to 13.0% of New Zealand respondents. • 82.1% of participants had a general practitioner. • Many other points mentioned were derived from qualitative subsections of the questionnaire. | Unnamed survey. Designed by research team | The questionnaire was developed by the research team in consultation with members of the trans community in Australia and New Zealand. | Yes |

(Continued)

Table 1. (Continued)

| Author and year, Study title, Study aim | Country and/or region | Population, Sample Size, Sample age | Methods | Access factors assessed | Overall findings | Instrument(s) used | Instrument(s) development | Inclusion of PPI |
|---|---|---|---|---|---|---|---|---|
| (Puckett et al., 2018) Study title: Barriers to Gender-Affirming Care for Transgender and Gender Nonconforming Individuals Study aim: 1) rates of pursuing or desiring to pursue different forms of gender-affirming healthcare (i.e., hormone therapy, top surgery, bottom surgery, puberty blockers); and 2) qualitative responses regarding barriers participants encountered in each of these areas. | USA | Trans/NB (n = 256) Age: 16+ | • Cross-sectional survey • Online | • Lack of service information • Pathways difficult to navigate • Experienced discrimination in healthcare • Family/peers not supportive • Age • Fertility/child related considerations • Not out • Denied care • Geographical considerations • Financial cost • Health insurance coverage/denial • Expertise and skill of provider (Need to teach) • Mental health assessment requirement • General health risk | • 61.3% were receive hormone therapy 22.7% had undergone top surgery and 5.5% had undergone bottom surgery • The most common reported barriers were financial in nature. • Many other points mentioned were derived from qualitative subsections of the questionnaire. | Unnamed survey. Designed by research team | The questionnaire was developed by the research team in consultation with members of the trans community in through a community advisory board. The group gave feedback on study questions, design, recruitment materials/ methods as well as discussions concerning the findings of the study as relative to their lived experiences. | Yes |
| (Reback et al., 2018) Study title: Health Disparities, Risk Behaviours and Healthcare Utilization Among Transgender Women in Los Angeles County: A Comparison from 1998–1999 to 2015–2016. Study aims: A comparison study of transgender women's health disparities, HIV risk behaviours, substance use, healthcare utilization, experiences of discrimination and HIV preva- lence over two distal time points. | USA, Los Angeles County | Trans women (n = 244 [study1] and 271 [study2]) Age: 18+ | • Cross-sectional survey (two time periods) • Online and paper | • Experienced discrimination in healthcare • Homelessness • Health insurance coverage/denial | • Findings demonstrated that participants in the latter study reported significantly higher access to healthcare insurance and prescription hormones. • The use of non-medically prescribed hormone therapy dropped from 36.1% to 9.9% from study 1 to 2. • However, participants in the latter study also reported lower levels of income; and, elevated prevalence of homelessness, HIV and lifetime STIs, receptive condomless anal intercourse with casual partner (s), and reported physical harassment/abuse. | The Los Angeles Transgender Health Survey | Developed by the first author and colleagues in 1997, in consultation with transgender women community members, and updated as community needs have changed, the Los Angeles Transgender Health Survey consists of seven modules: screening, sociodemographic characteristics, health care access and medical history including HIV services and hormone use/misuse, sexual behaviours (at all stages of gender transition) including HIV risk/ protective behaviours, substance use, legal and psychosocial issues including stigma and discrimination, and HIV prevention. | Yes |

(Continued)

Table 1. (Continued)

| Author and year, Study title, Study aim | Country and/or region | Population, Sample Size, Sample age | Methods | Access factors assessed | Overall findings | Instrument(s) used | Instrument(s) development | Inclusion of PPI |
|---|---|---|---|---|---|---|---|---|
| (Ross et al., 2021) **Study title:** **Experienced barriers of care within European treatment seeking transgender individuals: A multicentre ENIGI follow-up study** **Study aim:** To evaluate the experienced barriers of care for treatment-seeking trans individuals (TSTG) in three large European clinics. | Belgium, Netherlands, Germany | Trans (n = 307) Age: 17+ | • Cross-sectional survey • Online | • Gender identity considerations • Family/peers not supportive • Fear of seeking care • Age • Denied care • Geographical considerations • Financial cost • Comfort with provider • Mental health assessment requirement • Model of care (centralised MDT vs decentralised) • Need to prove gender • Mental health hx a barrier | • The majority of participants reported various EBOC, oftentimes more than one. • The most-frequently reported EBOCs pertained to the lack of family and friends' support (28.7%, n = 88) and travel time and costs (27.7%, n = 85), whereas around one-fifth felt hindered by treatment protocols. • Also, a significant share expressed the feeling that they had to convince their provider they needed care and/or express their wish in such way to increase their likelihood of receiving care. • A higher number of EBOCs reported was associated with more mental health problems, lower income and female gender. | ENIGI (European Network for the investigation of Gender Incongruence) | This study was initiated by the European Network for the Investigation of Gender Incongruence (ENIGI), which is a collaboration between four high-volume European clinics specialised in gender-affirming care, located in Amsterdam (the Netherlands), Ghent (Belgium), Hamburg (Germany) and Oslo (Norway). The participating clinics have applied similar assessment batteries available for research. No PPI involvement is evident from the paper. | No |
| (Sanchez et al., 2009) **Study title: Healthcare Utilisation, Barriers to Care, Hormone Usage Among Male to Female Transgender Persons in New York City** **Study aim:** To explore health care utilization, barriers to care, and hormone use among male-to-female transgender persons residing in New York City | USA, New York | Trans women (n = 101) Age: 18+ | • Cross-sectional survey • In person (Community resources) | • Homelessness • Financial cost • Expertise and skill of provider (Need to teach) • Comfort with provider | • Access to a provider knowledgeable about transgender health issues was the most reported barrier to care (32%) • This was followed by access to a transgender friendly provider (30%), cost (29%), location (18%) and language (13%). | Unnamed survey. Designed by research team | The questionnaire was developed by the research team. No specifics in paper on PPI contribution to project. | No |

(Continued)

**Table 1.** (Continued)

| Author and year, Study title, Study aim | Country and/or region | Population, Sample Size, Sample age | Methods | Access factors assessed | Overall findings | Instrument(s) used | Instrument(s) development | Inclusion of PPI |
|---|---|---|---|---|---|---|---|---|
| (Scheim et al., 2019) **Study title: Health care access among transgender and nonbinary people in Canada, 2021: a cross-sectional survey** Study aim: To examine access to care among trans and nonbinary people in Canada, and compares health care access across provinces or regions | Canada | Trans (n = 2217) Age: 14+ | • Cross-sectional survey • Online and in person | • Unsure if want/need medical intervention • Waiting times • Comfort with provider | • Of the 2217 trans and nonbinary respondents, most had a primary care provider (n = 1803; 81.4%. • Of those, 52.3%, had a primary care provider with whom they were comfortable discussing trans health issues, and 44.4% reported an unmet health care need. • Among participants who needed gender-affirming medical treatment (n = 1627), self-defined treatment completion ranged from an estimated 16.8% in Newfoundland and Labrador to 59.1% in Quebec. • Of those who needed but had not completed gender-affirming care at the time of the study 40.7% were on a wait-list, most often for surgery. | Trans Pulse | For the 2019 survey, the research team adapted core survey items from Trans PULSE Ontario, a province-wide study conducted by members of the research team in 2009–2010. The 2009 survey was developed by a 10-person community based research team and revised after reviews by the project's 16-member Community Engagement Team | Yes |
| (Sha et al., 2021) **Study title: Gender minority stress and access to health care services among transgender women and transfeminine people: results from a cross sectional study in China.** Study aim: The aim of this study was to examine the association between gender minority stress and access to specific health care services among transgender women and transfeminine people in China. | China | Trans women (n = 277) Age: 18+ | • Cross-sectional survey • Online and in person | • Experienced discrimination in healthcare • Not out • Internalised transphobia • Negative expectations for the future | • Overall, low uptake of gender affirming medical care, STI testing, PREP and PEP. • Discrimination and internalised transphobia are likely barriers to HIV and STI testing, though some gender minority stressors are also associated with higher uptake of some health care services. • Transgender people who experienced discrimination and internalised transphobia were more likely to use gender affirming hormones. | Unnamed survey. Designed by research team | The questionnaire was developed by the research team. No specifics in paper on PPI contribution to project. Partnering community based organisations shared the research. | No |

*(Continued)*

**Table 1.** (Continued)

| Author and year, Study title, Study aim | Country and/or region | Population, Sample Size, Sample age | Methods | Access factors assessed | Overall findings | Instrument(s) used | Instrument(s) development | Inclusion of PPI |
|---|---|---|---|---|---|---|---|---|
| (Sineath et al., 2016) Study title: Determinants of and Barriers to Hormonal and Surgical Treatment Receipt Among Transgender People Study aim: To explore current treatments and barriers to care | USA, Georgia | Trans/NB (n = 280) Age: 18+ | • Cross-sectional survey • Online | • Fear of seeking care • Unsure if want/need medical intervention • Age • Readiness • Finding a provider/ service • Financial cost • Health insurance coverage/denial • Expertise and skill of provider (Need to teach) • Mental health assessment requirement | • The respective percentages of ever and current HT were 58% and 47% for transwomen and 63% and 57% for transmen. • Having health insurance was not associated with GCT receipt. • Treatment cost was named as the main problem by 23% of transwomen and 29% of transmen. • Accessing a qualified healthcare provider for transgender-related care was listed as the primary reason for not receiving surgery by 41% of transmen and 2% of transwomen. | Unnamed survey. Designed by research team | The questionnaire was developed by the research team. No specifics in paper on PPI contribution to project. Partnering community based organisations shared the research. | No |
| (Sorbara et al., 2021) Study title: Factors Associated With Age of Presentation to Gender-Affirming Medical Care Study aim: to explore and compare care-seeking experiences of older and younger youth seeking GAMC, including assessment of both youth and caregivers. | Canada, Toronto | Trans youth/ caregivers (n = 121 youth & n = 121 caregivers) Age: <15 and >15 | • Cross-sectional survey • In clinic | • Gender identity considerations • Family/peers not supportive • Age • Not out • Uncomfortable discussing gender • Religious considerations (self or parents) | • Family environment appears to be a key determinant of when youth present to gender affirming medical care. • Compared with younger-presenting youth, older-presenting youth recognised gender incongruence at older ages, were less likely to have caregivers who helped them access care or LGBTQ+ (lesbian, gay, bisexual, transgender, queer) family members, more often endorsed familial religious affiliations, and experienced greater youth-caregiver disagreement around importance of GAMC. | Unnamed survey. Designed by research team | The questionnaire was developed by the research team. 2 clinic participants and 4 parents were involved in feedback during validation. | Yes |

*(Continued)*

Table 1. (Continued)

| Author and year, Study title, Study aim | Country and/or region | Population, Sample Size, Sample age | Methods | Access factors assessed | Overall findings | Instrument(s) used | Instrument(s) development | Inclusion of PPI |
|---|---|---|---|---|---|---|---|---|
| (Strauss et al., 2021) Study title: Perspectives of trans and gender diverse young people accessing primary care and gender-affirming medical services: Findings from Trans Pathways Study aim: To explore the experiences of trans young people accessing primary care and gender-affirming medical services in Australia for reasons related to their gender. | Australia | Trans/NB (n = 859) Age: 14–25 | • Cross-sectional survey • Online | • Age • Waiting times • Comfort with provider • Confidence in service available | • Most participants were aged 18 or older when they accessed gender affirming medical service (75.4%). • Private endocrinologist (57.3%) and private surgeon (35.2%) were the most accessed professionals. • 66.9% of youth found gender services highly or moderately satisfactory and 89% found staff respectful. • Qualitative barrier themes from write in section on questionnaire included: lost referrals, miscommunication, wait times, cost, travel, knowledge of HCP and lack of services. | Trans Pathway | The Trans Pathways survey utilised an online questionnaire developed jointly through community consultation with trans young people and parents of trans young people to assess mental health and experiences of health services among trans young people. | Yes |
| (Strousma et al., 2020) Study title: Insurance Coverage and Use of Hormones Among Transgender Respondents to a National Survey Study aim: We undertook a study to assess the associations between barriers to insurance coverage for gender-affirming hormones (either lack of insurance or claim denial) and patterns of hormone use among transgender adults. | USA | Trans/NB (n = 27,715) Age: 18+ | • Cross-sectional survey • Online | • Gender identity considerations • Health insurance coverage/denial | • Of 12,037 transgender adults using hormones, 992 (9.17%) were using non-prescription hormones. • Among insured respondents, 2,528 (20.81%) reported that their claims were denied. • Lack of insurance coverage for gender-affirming hormones is associated with lower overall odds of hormone use and higher odds of use of non-prescription hormones; such barriers may thus be linked to unmonitored and unsafe medication use, and increase the risks for adverse health outcomes | 2015 United States Transgender Survey (a sub section of overall findings) | The USTS survey instrument was developed over the course of a year by a core team of researchers and advocates in collaboration with dozens of individuals with lived experience, advocacy and research experience, and subject-matter expertise. Exact details on manner of PPI not specified but mentioned. A pilot study of 100 individuals influenced the survey too. | Yes |

(Continued)

**Table 1.** (Continued)

| Author and year, Study title, Study aim | Country and/or region | Population, Sample Size, Sample age | Methods | Access factors assessed | Overall findings | Instrument(s) used | Instrument(s) development | Inclusion of PPI |
|---|---|---|---|---|---|---|---|---|
| (Wilson et al., 2015) **Study title: Connecting the Dots: Examining Transgender Women's Utilization of Transition-Related Medical Care and Associations with Mental Health, Substance Use, and HIV** **Study aim: To examine Transgender Women's Utilization of Transition-Related Medical Care and Associations with Mental Health, Substance Use, and HIV** | USA, San Francisco | Trans women (n = 314) Age: 18+ | • Cross-sectional survey • Unknown | • Gender identity considerations • Age • Ethnoracial considerations • Health insurance coverage/denial • Mental health hx a barrier | • Disparities in accessing gender affirming care differ between ethnicities, especially among African American and Latina women. • Health insurance is a factor that impacts access • Whether transwomen identified as transgender or female impacted odds of accessing care. | Transfemales Empowered to Address Community Health (TEACH) study | The questionnaire was developed by the research team. No specifics in paper on PPI contribution to project. | No |
| (Polonijo et al., 2020) **Study title: Transgender and Gender Nonconforming Patient Experience in the Inland Empire, California** **Study aim: To examine physical health, mental health, health care access, and health care discrimination among TGNC individuals in California's Inland Empire.** | USA, California's Inland Island | Trans/NB (n = 90) Age: 18+ | • Cross-sectional survey • Online | • Experienced discrimination in healthcare • Age • Ethnoracial considerations • Finding a provider/ service • Denied care • Financial cost • Health insurance coverage/denial • Expertise and skill of provider (Need to teach) | • Less than half of all respondents reported it was very easy to find a physician (48.9%, n = 44) or mental health care professional (36.7%, n = 33) willing to provide routine care, with younger respondents reporting more difficulty. • More than three-quarters (76.7%, n = 69) of the total sample agreed there are "not enough health professionals adequately trained to care for people who are transgender," • 18.9% (n = 17) of respondents indicated that health care professionals refused to touch them or used excessive precautions, and this was more common among respondents assigned male at birth (31.4%, n = 11) than respondents assigned female at birth (12.2%, n = 6; p = 0.031). | Unnamed survey. Designed by research team | The questionnaire was developed by the research team. No specifics in paper on PPI contribution to project. Partnering community based organisations gave feedback on the design. | No |

(Continued)

Table 1. (Continued)

| Author and year, Study title, Study aim | Country and/or region | Population, Sample Size, Sample age | Methods | Access factors assessed | Overall findings | Instrument(s) used | Instrument(s) development | Inclusion of PPI |
|---|---|---|---|---|---|---|---|---|
| (Gandy et al., 2021) Study title: Trans*Forming Access and Care in Rural Areas: A Community-Engaged Approach Study aim: to use a stakeholder-engaged mixed-methods approach to begin to determine what TGD people, both adults and minors, require in a rural Appalachian American context such as West Virginia to have their health care needs met. | USA, West Virginia, Appalachian America | Trans/NB (n = 24) Age: 18+ | • Cross-sectional survey • Online | • Lack of service information • Family/peers not supportive • Unsure if want/need medical intervention • Fear of discrimination in healthcare • Geographical considerations • Financial cost • Health insurance coverage/denial • Expertise and skill of provider (Need to teach) • Comfort with provider | • Participants had to travel on average nearly an hour and a half (1.425 h) to access gender-related care. • 70% of participants had to travel out of state for gender-affirming care. • (20.8%) had received hormone replacement therapy and wished to have surgery but could not afford it. • Other barriers to accessing care included lack of provider trust (n = 3, 12.5%), fear of discrimination in employment and housing (n = 1, 4.2%), and lack of parental approval (n = 1, 4.2%). | Unnamed survey. Designed by research team | Researchers of this study engaged a community advisory board of TGD adults and parents of TGD minors who live in West Virginia throughout the research process, including the design, data collection, and analysis of results. Meetings were held monthly throughout the project. | Yes |
| (Tabaac et al., 2020) Study title: Barriers to Gender-affirming Surgery Consultations in a Sample of Transmasculine Patients in Boston, Mass. Study aim: to identify what barriers top and bottom surgery patients experienced before their initial consults and quantifying the out-of pocket costs spent on gender-affirming care | USA, Massachusetts | Trans masculine (n = 160) Age: 15+ | • Cross-sectional survey • In clinic | • Fear of seeking care • Age • Fertility/child related considerations • Readiness • Financial cost • Health insurance coverage/denial • Expertise and skill of provider (Need to teach) • Mental health assessment requirement | • The total number of barriers reported did not significantly differ by type of surgery, with an average of 3–4 total barriers among all participants. • The barriers most commonly reported as affecting surgery access were insurance coverage, cost of surgery, and getting one or more mental health letters. • Bottom surgery patients were significantly more likely than top surgery patients to report surgical readiness (P = 0.04) and hair removal (P = 0.002) as affecting access. | The transmasculine surgical expectations study | The questionnaire was developed by the research team. No specifics in paper on PPI contribution to project. | No |

(Continued)

**Table 1.** (Continued)

| Author and year, Study title, Study aim | Country and/or region | Population, Sample Size, Sample age | Methods | Access factors assessed | Overall findings | Instrument(s) used | Instrument(s) development | Inclusion of PPI |
|---|---|---|---|---|---|---|---|---|
| (Nolan et al., 2020) Study title: Barriers to Bottom Surgery for Transgender Men Study aim: To examines barriers to bottom surgery (phalloplasty and metoidioplasty) for 104 transgender men and nonbinary patients assigned female at birth who had not had this surgery. | USA, New York | Trans masculine/ NB (n = 104) Age: 18+ | • Cross-sectional survey<br>• In clinic | • Lack of service information<br>• Family/peers not supportive<br>• Fear of seeking care<br>• Finding a provider/ service<br>• Geographical considerations<br>• Financial cost<br>• Health insurance coverage/denial<br>• Expertise and skill of provider (Need to teach)<br>• Comfort with provider<br>• Mental health assessment requirement | • The most common. barrier affecting patients' decision to pursue or forgo bottom surgery was "financial barriers" (43.3 percent, n = 45).<br>• "Difficulty with my insurance" (26.9 percent, n = 28) and "difficulty finding a surgeon who takes my insurance" (13.5 percent, n = 14) were also important. | Unnamed survey. Designed by research team | The questionnaire was developed by the research team. No specifics in paper on PPI contribution to project. | No |
| (Nolan et al., 2020) Study title: Continued Barriers to Top Surgery among Transgender Men Study aim: To examines barriers to top/chest surgery | USA, New York | Trans masculine/ NB (n = 58) Age: 18+ | • Cross-sectional survey<br>• In clinic | • Lack of service information<br>• Family/peers not supportive<br>• Fear of seeking care<br>• Finding a provider/ service<br>• Geographical considerations<br>• Financial cost<br>• Health insurance coverage/denial<br>• Expertise and skill of provider (Need to teach)<br>• Comfort with provider<br>• Mental health assessment requirement | • In this study almost all patients had some insurance.<br>• The top consideration when choosing a surgeon was financial: whether the surgeon accepted the subject's insurance (80 percent).<br>• Fear of complications was moderately high in hindering decision to have surgery (31%) | Unnamed survey. Designed by research team | The questionnaire was developed by the research team. No specifics in paper on PPI contribution to project. | No |

**Table 2. Types of patient and public involvement.**

| Type of PPI described | Present in # studies | Referenced studies |
|---|---|---|
| Expert panel of gender diverse individuals | (n = 7) | [9, 22, 23, 37–41] |
| Consultation with gender diverse individuals | (n = 5 datasets, 9 studies) | [17–21, 31, 42–44] |
| Consultation with LGBT/Trans specific advocacy groups | (n = 3) | [40, 45] |
| Expert panel with parents of gender diverse youth | (n = 2) | [39, 46] |
| Consultation with parents of gender diverse youth | (n = 2) | [31, 44] |
| Expert panel with healthcare providers | (n = 1) | [22, 23] |

online and in-person methods. The recruitment strategy of one study was not clearly reported [58].

The sample sizes of the studies ranged from 24 to 27,715 participants, with larger responses obtained from a national American survey on discrimination. The median respondent rate was 278 respondents. The majority of studies focused on healthcare access, with a particular emphasis on hormonal care (n = 36). Five studies solely reported on access to surgical care [26–30].

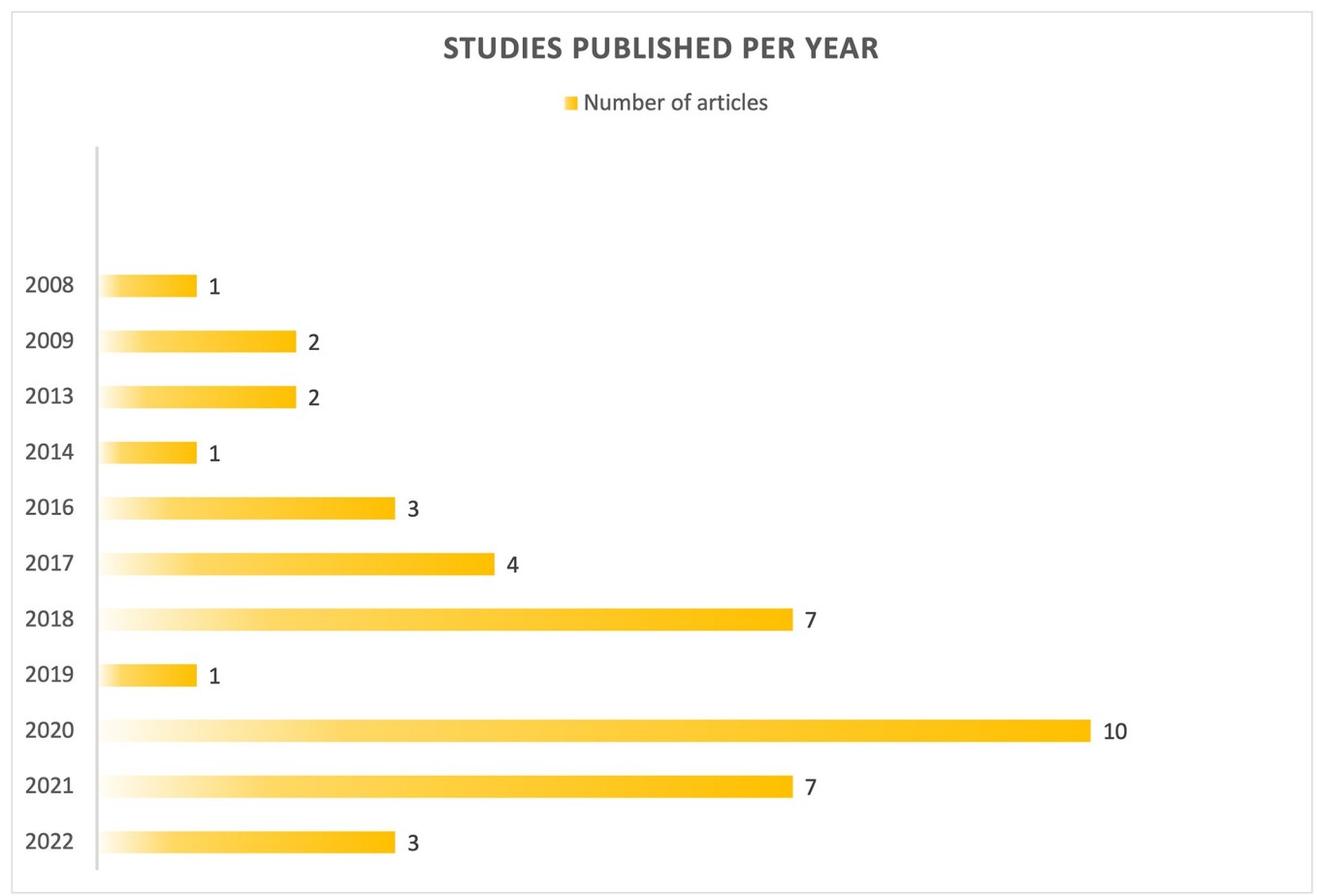

**Fig 2. Number of included studies by year.**

### 3.2 Theoretical underpinnings

The Minority Stress Theory (MST) is the most frequently cited theory in the literature (n = 9), with included studies employing this framework to design quantitative instruments. Even when not explicitly mentioned or stated, the influence of MST is apparent in the included studies. The theory posits that individuals belonging to minority groups experience higher levels of external/distal stressors, such as prejudice, rejection, and discrimination, which can lead to internal/proximal stressors, including concealment of one's identity, internalised homophobia/transphobia, and hypervigilance and anxiety related to prejudice/victimization. These factors can negatively affect health status and healthcare utilization. Therefore, many of the included articles focus on experiences of discrimination, harassment, victimization, peer support, and current mental health and self-esteem status.

In addition, one of the included studies noted the relevance of Ansara's cisgender theoretical framework [31], which examines cisgenderism and how clinicians often wrongly assume gender, and this causes negative experiences of clinical care [32, 33]. While healthcare access theories are infrequently referenced in the literature, one study [9] mentions Levesque's healthcare theory (2013) [7]. However, other healthcare access theories such as Pechanskay and Thomas (1981) [34], Andersen (1995) [35], or Ryvicker (2018) [36] are not mentioned. These healthcare access theories were considered for this review as a theoretical framework but Levesque's healthcare access theory was specifically chosen for this paper due to its structured approach and clear categories. This proved invaluable in the organization and analysis of a large data set for this scoping review. As the study aimed to explore a broad range of factors within the literature, the authors found that Levesque's framework provided a systematic and comprehensive foundation, enhancing the ability to discern and categorize various factors influencing healthcare access for transgender and non-binary individuals. The theory's consideration of individual and systemic factors, including socioeconomic status, cultural beliefs, and health literacy, allowed for a detailed exploration of the challenges faced by this population. Moreover, its recognition of the dynamic nature of healthcare access, influenced by policy changes, social norms, and healthcare delivery models, made it particularly suited for examining the evolving landscape of transgender healthcare.

### 3.3 Design and development of instrument (Patient and public involvement [PPI])

Patient and public consultation were common in the included studies, even though they were quantitative in nature. Almost half of the studies (n = 19) included some form of patient and public involvement. From these nineteen studies, five data sets were derived from the same American sample of patients [17–21], and two of the nineteen data sets came from the same German sample [22, 23]. Therefore, fourteen unique data sets included different PPI strategies as part of their quantitative methodologies (n = 14). These are described in Table 2.

The most common form of patient and public involvement was the development of an expert panel or advisory group of transgender and gender diverse individuals (n = 7) who met at timed intervals throughout the project. The next most common was consultation with transgender and gender diverse individuals (n = 5) and was normally not described as recurrent.

Caregivers, parents and LGBT advocacy groups were sparingly utilized for their expertise and healthcare providers were the most underrepresented as advisory stakeholders (n = 1).

### 3.4 Factors influencing healthcare navigation

The present study has identified a total of forty-one distinctive factors that can either facilitate or impede healthcare access to gender care based on a comprehensive review of the included

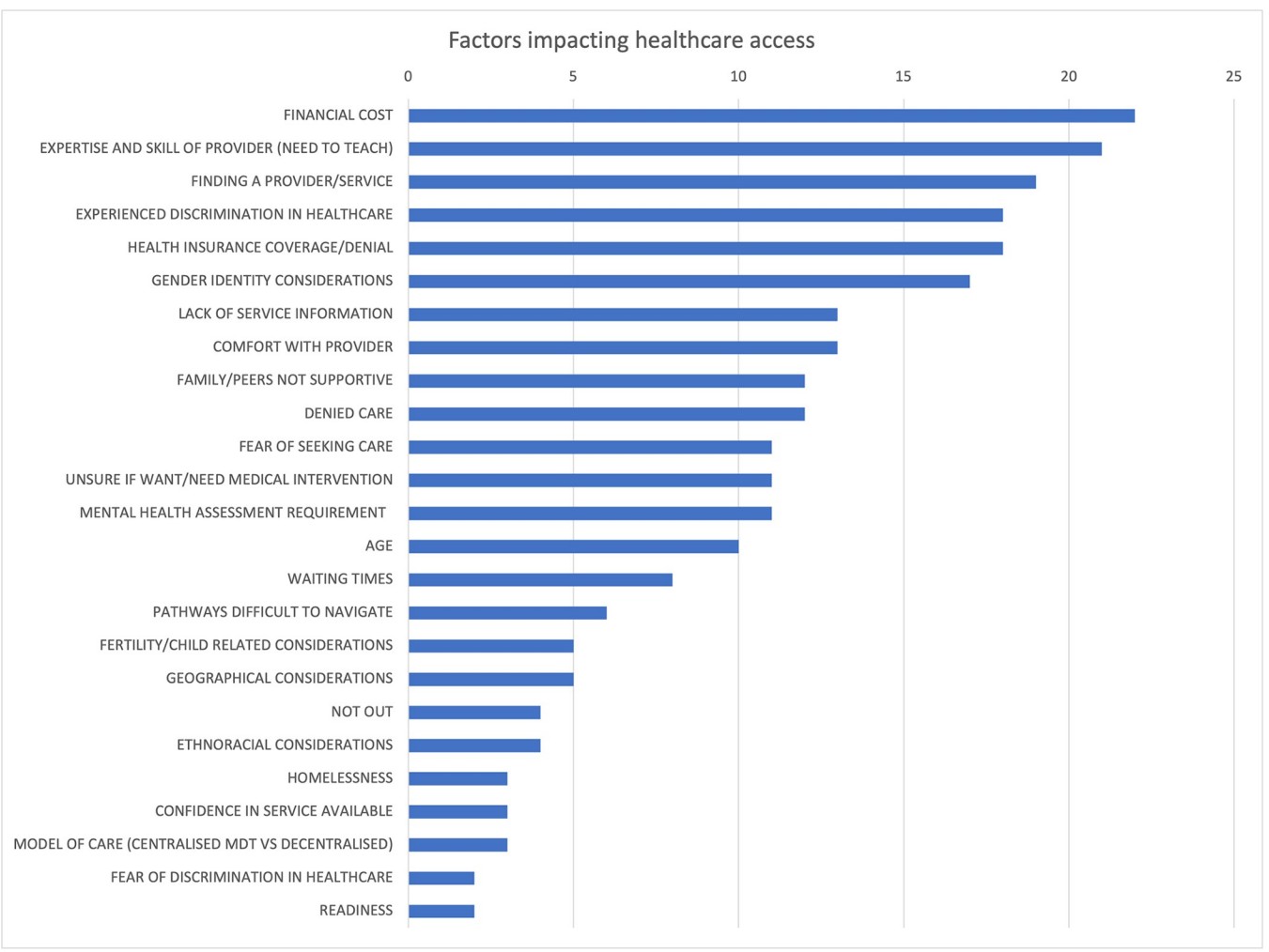

**Fig 3. 25 most referenced access factors.**

studies. These factors have been systematically classified and charted according to Levesque's (2013) five dimensions of accessibility, namely: 1) approachability, 2) acceptability, 3) availability and accommodation, 4) affordability, and 5) appropriateness. A detailed account of the charting and categorisation of these factors is provided in S1 Data. All studies were included for the initial analysis. Notably, this paper has highlighted the top 25 factors which are presented in Fig 3.

Regarding approachability, the most significant factor identified was the lack of service information, as indicated in thirteen of the studies [25–29, 38–39, 41, 46–50]. Transgender and gender diverse individuals encountered difficulties in sourcing accurate information about service availability and the available interventions. Furthermore, the pathways to access these services were frequently perceived as difficult to navigate or comprehend, as evidenced by (n = 6) studies [26, 27, 38, 41, 47, 48].

Acceptability is determined by cultural and societal factors that dictate whether people are willing to accept certain aspects of care and whether seeking care is deemed appropriate for them. This category prompted the most factors related to accessing care. The present study examined the impact of previous experiences of discrimination on healthcare-seeking

behaviours among a sample of individuals across eighteen studies [20, 24, 26, 37–38, 40–43, 45, 47, 48, 50–55]. Findings indicate that discriminatory behaviours, such as invasive questioning, denial of care, incorrect name and pronoun use, and inappropriate physical exams, were associated with decreased likelihood of seeking future healthcare needs.

Furthermore, the study identified differences in the ease of access to care among gender identity groups in seventeen studies [17–21, 23–25, 37, 38, 44, 45, 48, 49, 56–58], with distinct disparities observed between transfeminine and transmasculine individuals and between binary and non-binary healthcare seekers. One notable disparity highlighted in the study is that a higher proportion of binary individuals demonstrated greater access to healthcare compared to their non-binary counterparts. Additional disparities were observed, including variations in access to surgery between transmasculine and transfeminine individuals, as well as discrepancies in access related to ethnicity or race. These results suggest that cultural and societal values and norms may profoundly influence healthcare behaviour across these groups. Importantly, this dimension highlights the critical role of support and acceptance from family and peers in influencing healthcare access in twelve of the included studies [25, 26, 28, 29, 37, 39, 41, 44–46, 49, 57].

In the context of healthcare seeking and reaching, availability and accommodation emerged as key factors, with the most prominent challenge being the identification of a provider or service that was willing to prescribe or treat (n = 19) [20, 23, 25, 26, 28, 29, 37, 38, 40, 42, 45, 47–50, 52, 55, 56, 59]. Notably, a significant number of participants reported being denied care during the process of seeking healthcare (n = 12) [18, 20, 25, 26, 38, 40, 41, 45, 51, 52, 55, 57], and persistent barriers to accessing care were associated with long waiting times (n = 8) [9, 31, 40, 42, 46, 48, 50, 51]. One study explored homelessness as a factor impacting access to care [60].

The present cohort of patients considered affordability as a crucial determinant for accessing healthcare services. Financial cost emerged as the most frequently cited factor impacting access (n = 22) [18, 20, 25, 26, 28–30, 38, 39, 41–43, 45, 47–50, 52, 55, 56, 59, 61]. Financial factors emerged as a predominant concern in studies conducted in the United States, reflecting the American context, where most studies were based. In contrast, studies conducted in European settings, highlighted waiting times and the appropriateness of services as prominent factors influencing healthcare access for transgender and non-binary individuals. Issues related to insurance coverage and denial were commonly examined in the included studies (n = 18) [17, 20, 21, 26, 28–30, 37, 39, 41–43, 48, 52, 55, 56, 58, 59].

Lastly, the appropriateness of services was evaluated based on the congruence between services provided and the client's needs, as well as the quality of care received, including the interpersonal aspects of the care process. Findings from the studies revealed that participants frequently perceived healthcare professionals as lacking adequate skills or knowledge to deliver care, resulting in the patients needing to educate their providers (n = 21) [20, 26–30, 38–42, 45, 46, 48, 50–52, 55, 56, 59, 61]. Furthermore, a considerable number of participants viewed mental health assessments as a redundant barrier to care (n = 11) [22, 25, 28–30, 40, 41, 47, 52, 57, 59].

### 3.5 Youth and accessing gender related healthcare

Eight studies were included in this analysis, involving participants under the age of 16, with the youngest child being 4.7 years old [9, 30, 31, 44, 46, 49, 51, 53]. The majority of studies originated from North America, with three from the United States and three from Canada. One study each was conducted in Mexico and Australia.

Of the Canadian studies, one cross-sectional study focused on healthcare access and comfort of care in primary care and gender services, which recruited individuals aged 14 and

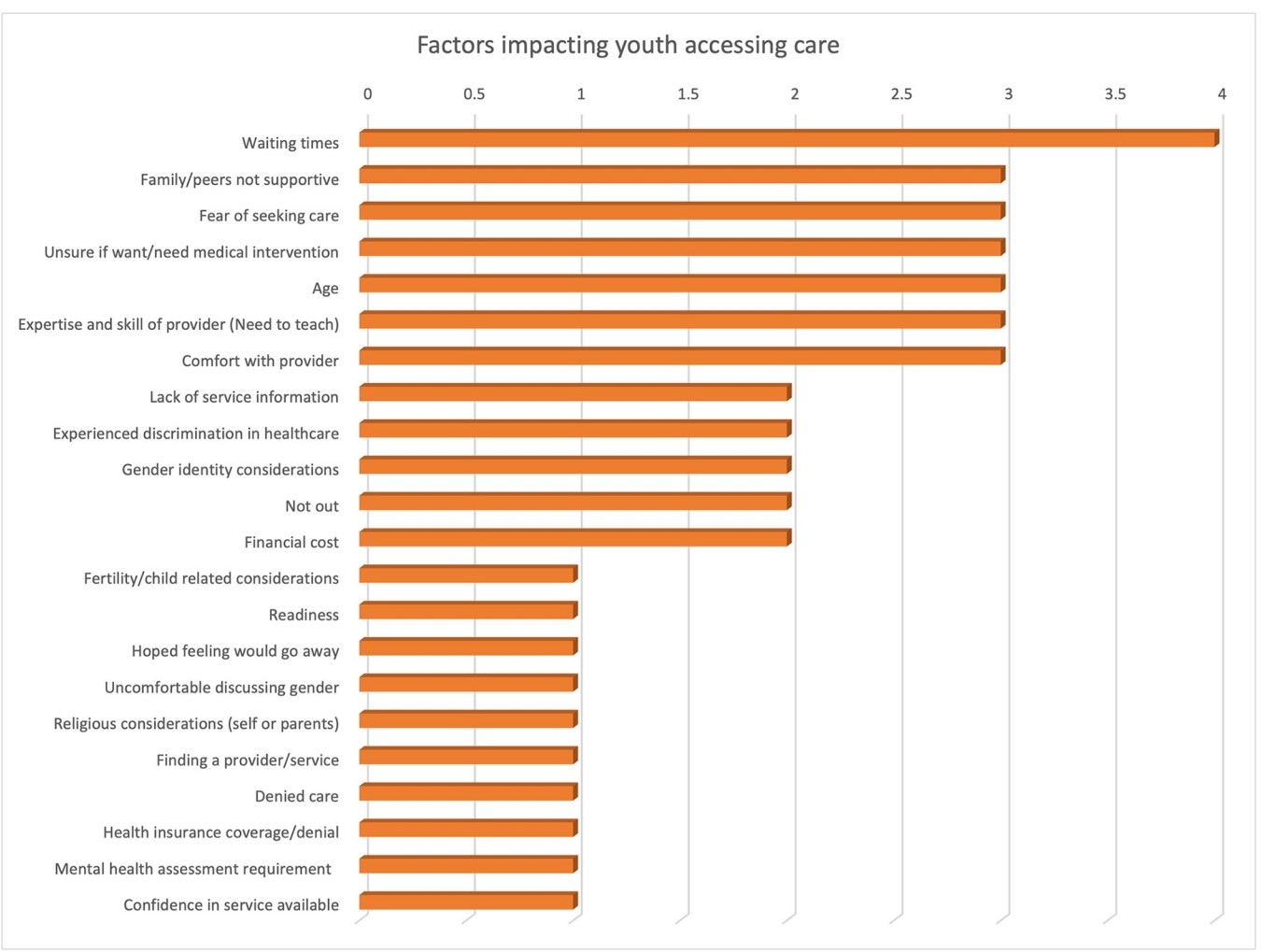

**Fig 4. Access factors among youth.**

above. The remaining two Canadian studies recruited through interdisciplinary gender clinics specializing in gender care for young people. Of the American studies, one study recruited from a specialized gender clinic in Manitoba, while another study assessed the needs and experiences of non-binary youth through a cross-sectional study.

One study focused on access to top surgery in Boston and included participants aged 15 and above. Lastly, one study focused on discrimination and violence in Mexico and included children aged 14 and above, while one Australian study assessed the perspectives of transgender and gender diverse young people accessing gender care between the ages of 14 and 25.

A total of 22 unique factors were identified as issues facing youth in accessing gender care (See Fig 4). A detailed account of the charting and categorisation of these factors is provided in S1 Data. Waiting times were the most commonly referenced issue, highlighting concerns around availability and accommodation. Acceptance and support from family and peers were identified as key factors in the acceptability dimension. Feelings of fear and uncertainty were also commonly reported, with some youth unsure whether they wanted or needed medical interventions to affirm their gender. The age of seeking care was noted as an important factor, and the challenges of finding competent providers persisted across all age groups.

Although financial cost and affordability concerns were mentioned, they were not considered to be as significant as other issues. Nevertheless, opinions regarding appropriateness were frequently expressed, with many young people and parents commenting on the lack of provider expertise and ability to meet their needs.

The authors observed that non-binary identities and gender diversity were more prevalent among younger cohorts, and that there were varying models of care, with specialised gender clinics playing a crucial role in providing care.

## 4. Discussion

Access to gender-affirming care is a critical issue for transgender people, who often face significant barriers to accessing the care they need. This is the first scoping review to compare the quantitative instruments used to assess healthcare access among transgender and gender diverse individuals across 32 individual datasets in 41 papers.

In order to provide a more comprehensive understanding of these findings, our study applied identified factors to Levesque et al.'s (2013) five dimensions of healthcare access: approachability, acceptability, availability and accommodation, affordability, and appropriateness.

Approachability refers to the ability of a person to identify and reach a healthcare service. Our review found that transgender people may not be aware of the healthcare services available or may lack accurate information about their options for gender-affirming care. To improve approachability, healthcare providers and policymakers should work to increase awareness and education around available services and provide clear and accurate information about the benefits and risks of different treatment options.

None of the studies included in this review assessed health literacy, which the authors believe would be a novel area of investigation for future research. Furthermore, examining the impact of co-occurring diagnoses on approachability is not fully understood, and should be a priority of future research.

Acceptability refers to the ability of a person to seek care without experiencing discrimination or judgment. Our review found that transgender people may face significant barriers to seeking care, including fear of discrimination or stigma, lack of trust in healthcare providers, and lack of support from parents and peers. A supportive family is particularly important to younger patients.

To improve acceptability, healthcare providers can create welcoming and supportive environments for transgender patients and work with families in increasing understanding of gender related needs. In addition, policymakers can work to address discrimination and stigma in the healthcare system. Given the current political climate in some parts of the world, where lawmakers are enacting policies that restrict healthcare access for transgender individuals, it is likely that the acceptability dimension of healthcare access may be threatened.

In light of these findings, further research on protective factors, and in particular the role of family/peer support as enablers to accessing healthcare services, would be useful. Furthermore, an examination of the role of internalised transphobia in deferring seeking care would be an interesting research aim for future studies.

Availability and accommodation refer to the ability of a person to find and access healthcare services. Our review found that transgender people may face geographic or financial barriers to accessing gender-affirming care, particularly in areas where there are limited healthcare providers or services. To improve availability and accommodation, policymakers can work to ensure that healthcare services are available and accessible to all individuals, regardless of their location or financial status. This may include considerations of various models of care and the

manner in which healthcare services are provided, which may differ based on the level of national resources available and political favour in funding services.

In countries where lengthy waiting lists represent a significant barrier to accessing healthcare services, it may be advantageous to introduce pilot interventions that provide information and support to prospective clients and their families while they wait to be seen. Additionally, given the growing interest in telehealth as a means of delivering gender services, it would be valuable to assess provider and patient perspectives on this approach.

Affordability refers to the ability of a person to pay for healthcare services. For transgender people living in regions were insurance coverage is needed to access healthcare, affordability can be a significant barrier. To improve affordability, healthcare providers and policymakers can work to ensure that gender-affirming care is affordable and accessible to all, regardless of ability to pay. This review found an oversaturation of research on the influence of cost and insurance on access to hormones. However, there is a dearth of research on experiences accessing surgical interventions and the effect of affordability on accessing surgery.

Appropriateness refers to the ability of a person to receive care that meets their individual needs and preferences. Our review found that transgender people may face significant barriers to engaging in the healthcare system for a variety of reasons, including lack of trust in healthcare providers, and mandatory mental health assessments. To improve appropriateness, healthcare providers can work to create individualised treatment plans that meet the unique needs and preferences of transgender patients while ensuring that the quality of clinical care is not jeopardised.

Furthermore, the review identifies opportunities for specialised healthcare services and providers to develop and implement training programs aimed at enhancing their proficiency in delivering gender-affirming care, as well as providing training to primary care providers to augment their knowledge in this area.

Patient and public involvement (PPI) is recognized as an advisory component in the conduct of research involving the transgender community. However, this review indicates that there is a dearth of detailed descriptions of the PPI processes employed, as well as a lack of information regarding sampling. While the study revealed that members of the transgender community were primarily involved in co-design, the inclusion of other key stakeholders was infrequent. It is recommended that future research incorporate nuanced PPI approaches to enhance the quality of research and report their involvement in a transparent manner.

The studies included in this review describe a diversity of gender affirming care access points, ranging from primary care providers, specialised gender clinics, telehealth services, or via non-prescribed methods such as obtaining medications from friends. This diversity of pathways poses challenges to research analysis, as there is no universal standard for accessing gender-affirming care. In addition, the differences in approach also makes it difficult to generalise results or experiences outside of the country and context that they are conducted in.

The review highlights that primary care access is more frequently observed in North America for adults, while gender clinics are more prevalent in Europe. Specialised gender clinics are more common for children and teenagers across the world. This poses an interesting question as to why the model of care changes so drastically from adolescence to adulthood when the complexity of need likely remains unchanged.

## 5. Strengths and limitations

This paper has several strengths that enhance its contribution to the field. Firstly, it provides a comprehensive overview of the key factors influencing healthcare access for transgender and

gender diverse individuals. Secondly, the study utilises an established healthcare access theory and systematically applies it to this population.

Additionally, the research compares the factors influencing healthcare access in adult cohorts versus youth cohorts, which adds a valuable perspective to the findings. The study team included medical and nursing professionals, demonstrating a united approach to the research objective. The clinicians involved have expertise in gender care, and an expert panel of transgender and gender diverse youth was consulted throughout the project, further enhancing the study's rigour and relevance.

However, this review has noteworthy limitations. Firstly, we employed a systematic approach to searching and screening relevant publications in four databases for this review and through grey literature searches, but our results are limited by the sensitivity of our search strategy and databases.

Secondly, the included research primarily represents the perspectives of transgender and non-binary individuals, with little input from clinicians who specialise in gender care or family members who may be important stakeholders in accessing care for individuals. Future research would benefit from examining the perspectives of multiple stakeholders. Additionally, this review does not provide a clear definition of what constitutes appropriate care and whether this definition varies between healthcare professionals, patients, and their families.

Thirdly, most of the studies included in this review utilized online samples, which could result in selection bias and non-representative samples, despite the benefit of large sample sizes.

Lastly, youth were underrepresented in this sample, with only eight papers including their perspectives. We conducted a sub-analysis of factors identified from youth-only studies, but the results are limited by the small number of studies and small sample sizes. It would be valuable to compare the factors influencing healthcare access between youth and adults in a specific geographic context, with a focus on the different aspects of healthcare access as described by Levesque.

## 6. Conclusion

In conclusion, this scoping review highlights the importance of Levesque et al.'s (2013) theory of access to healthcare in understanding the barriers faced by transgender people in accessing gender-affirming care. By examining the abilities of transgender people to perceive, seek, reach, pay, and engage with healthcare services, this review provides important insights into the factors that influence access to gender-affirming care for transgender people.

Though transgender and non-binary representation has increased in the media, there still exists many obstacles to accessing care in each dimension of Levesque's model. The findings of this review provides recommendations for practice and policy, and could be used to inform the development of interventions and policies that address the barriers faced by transgender people in accessing gender-affirming care. Ultimately, by improving access to gender-affirming care, healthcare providers and policymakers can help to improve the health and well-being of transgender people and ensure that they receive the care they need to live healthy and fulfilling lives.

## Supporting information

**S1 Checklist. Preferred Reporting Items for Systematic reviews and Meta-Analyses extension for Scoping Reviews (PRISMA-ScR) checklist.**
(DOCX)

**S1 Data. Charting and categorisation of all access factors.**
(XLSX)

## Author Contributions

**Conceptualization:** Seán Kearns.

**Data curation:** Seán Kearns, Philip Hardie.

**Formal analysis:** Seán Kearns, Philip Hardie, Karl Neff.

**Investigation:** Philip Hardie.

**Methodology:** Seán Kearns.

**Project administration:** Seán Kearns.

**Resources:** Seán Kearns.

**Software:** Seán Kearns.

**Supervision:** Philip Hardie.

**Validation:** Seán Kearns, Philip Hardie.

**Visualization:** Seán Kearns.

**Writing – original draft:** Seán Kearns.

**Writing – review & editing:** Seán Kearns, Philip Hardie, Donal O'Shea, Karl Neff.

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
