## [Decision Letter · Decision Letter 0]

22 Jan 2024

PONE-D-23-20695Instruments used to assess gender-affirming healthcare access: A scoping reviewPLOS ONE

Dear Dr. Kearns,

Thank you for submitting your manuscript to PLOS ONE. After careful consideration, we feel that it has merit but does not fully meet PLOS ONE’s publication criteria as it currently stands. Therefore, we invite you to submit a revised version of the manuscript that addresses the points raised during the review process.

We look forward to receiving your revised manuscript.

Kind regards,

Joseph Adu, PhD, MSc., Mphil

Academic Editor

PLOS ONE

Journal Requirements:

2. Please include your tables as part of your main manuscript and remove the individual files. Please note that supplementary tables (should remain/ be uploaded) as separate "supporting information" files

**Comments to the Author**

1. Is the manuscript technically sound, and do the data support the conclusions?

Reviewer #1: Yes

Reviewer #2: Yes

2. Has the statistical analysis been performed appropriately and rigorously? 

Reviewer #1: Yes

Reviewer #2: N/A

3. Have the authors made all data underlying the findings in their manuscript fully available?

Reviewer #1: Yes

Reviewer #2: Yes

4. Is the manuscript presented in an intelligible fashion and written in standard English?

Reviewer #1: Yes

Reviewer #2: Yes

**5. Review Comments to the Author**

**Reviewer #1 **

The authors have completed a rigorous scoping review on a very pertinent topic relevant for international health care and health systems. Access to healthcare for transgender and non-binary people is extremely important and this review demystifies the processes in how this population engage with health services and are met by healthcare professionals when seeking care. All revisions and queries are minor seeking to clarify certain aspects and exemplify the work done to readers. In-depth and meaningful engagement with the identified articles is evident and is a significant addition to the existing evidence-base. I particularly appreciated Figure 1 which included identification of studies using different methods, which is often not reported in a clear way in scoping reviews.

Abstract/Introduction/Title

1. The background section could include a statement on the unique nature of the review and its implications (i.e. being the first of its kind) which is mentioned later in the Conclusion.

2. Regarding choosing Levesque’s theory, could you mentioned otherst that were considered and why they did not fit with the review question/analysis approach?

3. Line 73 – remove “e.g.” from reference

4. Mention “university” librarian in the Abstract section

5. Transgender and non-binary representation in media is mentioned in Conclusion but not in background. I would suggest making a brief note about this in the Introduction to set the context.

6. Suggestion that the scope of the review and implications is broader than the “instruments” mentioned in the title. Based on title, I expected article to focus strictly on evaluation/assessment tools.

Methods

1. Line 107 - Were the healthcare access experts based in Ireland or international?

2. Line 136 – Were any further empirical articles found using Google Scholar as peer-reviewed articles are found this way too?

3. Line 146 – Use past tense i.e. “arose”

4. Line 146 – Was the criteria adapted after initial pilot?

5. State the third reviewer was also involved in full-text screening.

6. Line 163 – What format were results shared with panel e.g. email, group discussion?

Results

1. Line 174 - PRISMA flowchart appears to be a screen grab from Word with spellchecked underlined words. Suggest included clean version.

2. Line 192 – I would include reference for this study.

3. The research question focuses on quantitative instruments. The results includes studies with PPI which is commonly a qualitative approach. Did these studies have quantitative elements?

Discussion

1. Suggestion of including separate paragraph that details the future policy, practice and research recommendations as a result of your review. Not required as also works with the threaded throughout Discussion.

**Reviewer #2**

Overall, this paper contributes to an underdeveloped area regarding gender affirming healthcare and makes an important contribution to advocating for improved services and systems. I appreciated the writing style is concise and easy to read. I will offer some suggestions for further strengthening the paper below:

The authors clearly identify their use of Levesque’s healthcare access theory; I encourage them to consider introducing the model more within their introduction and reason for selection. For example, they mention the model recognizes socioeconomic status, cultural beliefs, and health literacy. Some examples or further expansion on the relevance of these may be beneficial for the reader to apply to this population.

Section 2.3 provides some details on the search and while the protocol is published elsewhere, it would be beneficial to provide some examples of search terms related to the concepts searched.

I think it would be beneficial if the authors address the rationale for a grey literature search, why this was included and what this would add?

The authors mention sharing results with a pre-existing expert panel consisting of transgender and non-binary youths, I suggest the authors considering locating themselves within this work and analysis and how their own experiences or backgrounds may have contributed to shaping this work. I am also curious if the author’s could clarify why the group consisted of youths given this research does not have an age restriction or limitation specified.

Line 271 mentions “disparities observed between transfeminine and transmasculine individuals and between binary and non-binary healthcare seekers” but no mention of what these might include, some clarification here would be beneficial for a reader.

Line 287 mentions “the present cohort of patients considered affordability as a crucial determinant for accessing healthcare services”, have the authors noted any difference here in terms of location and different healthcare coverage systems? Given the US is most highly representative I wonder if there were any differences to note.

Overall, I agree this article is well written and offers insights for better understanding healthcare access for transgender and non-binary individuals.

6. PLOS authors have the option to publish the peer review history of their article (what does this mean?). If published, this will include your full peer review and any attached files.

Reviewer #1: **Yes: **Andrew Darley

Reviewer #2: No

---

## [Author Response · Author response to Decision Letter 0]

23 Jan 2024

See attached file labelled response to reviewers

---

## [Decision Letter · Decision Letter 1]

31 Jan 2024

Instruments used to assess gender-affirming healthcare access: A scoping review

PONE-D-23-20695R1

Dear Sean,

We’re pleased to inform you that your manuscript has been judged scientifically suitable for publication and will be formally accepted for publication once it meets all outstanding technical requirements, including the journal referencing style.

Kind regards,

Joseph Adu, PhD, MSc., MPhil

Academic Editor

PLOS ONE

---

## [Editor Report · Acceptance letter]

22 Mar 2024

PONE-D-23-20695R1 

PLOS ONE

Dear Dr. Kearns, 

I'm pleased to inform you that your manuscript has been deemed suitable for publication in PLOS ONE. Congratulations! Your manuscript is now being handed over to our production team.

Kind regards, 

on behalf of

Dr Joseph Adu 

Academic Editor

PLOS ONE